

1    **MULTI RADAR PERFORMANCE IN THE MIDWESTERN UNITED STATES AT LARGE RANGES**

Micheal J. Simpson[1], Neil I. Fox[2]

[1]University of Missouri, School of Natural Resources, Water Resources Program, Department of
Soil, Environmental, and Atmospheric Sciences, 203-T ABNR Building, Columbia, Missouri, USA,
65201. Tel: +001 5857604031 Email: mjs5h7@mail.missouri.edu

[2]University of Missouri, School of Natural Resources, Water Resources Program, Department of
Soil, Environmental, and Atmospheric Sciences, 332 ABNR Building, Columbia, Missouri, USA,
65201. Tel: +001 5738822144 Email: FoxN@Missouri.edu

*Correspondence to*: Micheal J. Simpson (mjs5h7@mail.missouri.edu)
**Abstract.** Since the advent of dual-polarized technology, many studies have been conducted to determine
the extent to which the differential reflectivity (ZDR) and specific differential phase shift (KDP) add
benefits to estimating rain rates (R) to reflectivity (Z). It has been previously noted that this new
technology provides significant improvement to rain rate estimation, but only for ranges within 125 km
from the radar. Beyond this range, it is unclear as to whether the National Weather Service conventional
R(Z)-Convective algorithm is superior, as little research has investigated radar precipitation estimate
performance at large ranges. The current study investigates the performance of three radars, St. Louis
(KLSX), Kansas City (KEAX), and Springfield (KSGF), MO, with respect to range, with 15 terrestrial-
based tipping bucket gauges served as ground-truth to the radars. Over 1100 hours of precipitation data
were analyzed for the current study. It was found that, in general, performance degraded with range
beyond, approximately, 150 km from the radar. Probability of detection in addition to bias values
decreased, while the false alarm ratios increased as range increased. Bright-band contamination was
observed to play a potential role as large increases in the absolute bias and overall error values near 120
km for the cool season, and 150 km in the warm season. The analyses found further our understanding in
the strengths and limitations of the Next Generation Radar system overall, and from a seasonal
perspective.



## 1 Introduction

In 2012, the National Weather Service (NWS) began upgrading the Next Generation Radar

(NEXRAD) system from single- to dual-polarization. The potential benefits of this upgrade were
investigated by the National Severe Storms Laboratory (NSSL) and the Cooperative Institute for
Mesoscale Meteorological Studies. These advantages include, but are not limited to, (1) significant
improvement in radar rainfall estimation (Ryzhkov et al., 2005; Gourley et al., 2010) through better
representation of precipitation shape (Brandes et al., 2002; Gorgucci et al., 2000, 2006), (2)
discrimination between solid and liquid precipitation (Zrnic and Ryzhkov, 1996), allowing for better
distinction between areas of heavy rain and hail (Park et al., 2009; Giangrande and Ryzhkov, 2008;
Cunha et al., 2013),  (3) identifying the melting layer position in the radar field (Straka et al., 2000; Park
et al., 2009), and (4) calculating drop-size distributions retrieved from measurements of reflectivity (Z),
differential reflectivity (ZDR), and specific differential phase shift (KDP) as opposed to using ground-
based point located disdrometers (Zhang et al., 2001; Brandes et al., 2004; Anagnostou et al., 2008).

Despite the advantages listed above, there are several sources of uncertainty and challenges that

meteorologists and hydrometeorologists currently endure. For example, in order to ensure accuracy in
rain-rate (R) estimates, Ryzhkov et al., (2005) stated the (mis)calibration effects should, approximately,
be limited between $\pm1$ dBZ in reflectivity, and $\pm0.2$ dB for differential reflectivity. The specific
differential phase has been shown the be unaffected by beam blockage and other absolute calibration
issues (Zrnic and Ryzhkov, 1999), yet attenuation effects may be amplified at X-band radars where the
wavelength of the radar signal is more affected by the size of the hydrometeors (Delrieu et al., 2000;
Berne and Uijlenhoet, 2005).

Rain rate retrieval by weather radars is an estimation based upon the dielectric properties of the

hydrometeors encountered in the atmosphere. Therefore, there is no direct measurement of rainfall, and
this inherently introduces error. Although dual-polarized technology allows for the measurements of not



only Z, but also ZDR and KDP, conflicting studies have been conducted as to whether dual-polarized

radar rain rate algorithms have improved estimates over single-polarized radar rain rate algorithms. For

example, Gourley et al. (2010) and Cunha et al. (2015) reported that conventional R(Z) algorithms have

significantly better bias than algorithms containing ZDR and/or KDP, while others (e.g., Ryzhkov et al.,

2013; Simpson et al., 2016) report the opposite. This could be due, at least in part, to the fact that

hydrometeor types (e.g., rain versus hail) vary on spatial scales that cannot be easily resolved by even

densely gauged networks.

Multiple studies have found that, in general, the performance of radar rain rate estimates decrease

as range increases (Smith et al., 1996; Ryzhkov et al., 2003) which is caused, primarily, by degradation of

beam quality and broadening of the beam with range. Furthermore, the researchers also discuss how the

probability of detection at larger ranges decreases, as the radar beam overshoots shallow, stratiform

precipitation, including winter storms. Bright-banding can also play a crucial role in significantly

increasing the amount of precipitation estimated by the radar.

Despite these overall disadvantages, studies have shown that radar rainrate algorithms seldom

exceed absolute errors on the order of 10 mm h$^{-1}$. However, many of these studies have looked at a small

sample of rain events (on the order of 10-50 hours) (Kitchen and Jackson, 1993; Smith et al., 1996;

Ryzhkov et al., 2003; Gourley et al., 2010; Cunha et al., 2013). Additionally, few studies (e.g., Smith et

al., 1996; Cunha et al., 2015; Simpson et al., 2016) quantified meteorologically significant statistical

measures including the probability of detection and false alarm ratio. In order to get a better

understanding of the performance of weather radars on rain rate estimates, more data must be collected

over a broad range of precipitation regimes in addition to an overall broad region of interest.

The overarching objective of the current study was to assess the overall performance of three

different radars within the state of Missouri at various ranges from the radar, using terrestrial-based

tipping bucket gauges as ground-truth data. Radar rain rate estimation algorithms include 55 algorithms



encompassing standard R(Z) relations, in addition to algorithms containing dual-polarization variables
including ZDR and KDP. A rain rate echo classification algorithm was also tested for performance in
correctly identifying the suitable rain rate algorithm to choose based on the Z, ZDR, and KDP radar
fields. The current work expands upon that of Simpson et al. (2016) such that a larger sample of data were
analyzed (over 1000 hours of rainfall data from forty-six separate days in 2014) to encompass multiple
different precipitation regimes for both summer and winter, with several ground-truth tipping buckets to
analyze the performance of three separate radars at varying ranges, and further expanding upon the effects
of erroneous precipitation estimates on the overall radar error. Objectives for this study included, (1)
statistically analyze the performance of each radar at various ranges (compared against the terrestrial-
based gauges), (2) compute (a) the amount of precipitation incorrectly estimated by the radar (quantifying
the probability of false detection) and (b) the amount of precipitation incorrectly missed by the radar but
measured by the rain gauge, (3) test the overall best radar rain rate algorithm, and (4) perform objectives
(1), (2), and (3) while the data is separated into warm and cool seasons.

## 2   Study area and methods

### 2.1  Study area

National Weather Service radars from St. Louis (KLSX), Kansas City (KEAX), and Springfield

(KSGF), MO are able to scan the majority of the state of Missouri. Because of this, the three
aforementioned radars were used to assess overall performance in estimating precipitation for this study.
Each radar covered a 200-km radius for which a different number of gauges were within the domain:
KLSX, KEAX, and KSGF covered 9, 8, and 5 gauges, respectively (Figure 1).

Missouri is characterized as a continental type of climate, marked by relatively strong seasonality.

Furthermore, Missouri is subject to frequent changes in temperature, primarily due to its inland location
and its lack of proximity to any large lakes. All of Missouri experiences below-freezing temperatures on a



yearly-basis. For example, the majority of the state experiences, on average, 110 days with temperatures
below freezing, while the Bootheel (i.e., southeast region) registers, on average, 70 days of below
freezing days. This elaborates upon the typical northwest to southeast warming pattern of temperatures
observed in the state. Because of the large variability in temperature, the warm and cool seasons were
defined from an agronomic perspective, primarily taking probabilities of freezing into account. Based on
the climatological averages of Missouri, from 1983 to 2013, November through April registered average
minimum temperatures below freezing, and was considered the cool season, while May through
October's minimum average temperature were above freezing and constituted the warm season.

### 2.2 Rainfall data

Terrestrial-based (ground-truthed) precipitation gauge data were collected from 15 separate
weather stations within the Missouri Mesonet, established by the Commercial Agriculture Program of
University Extension (Table 1). All precipitation data were recorded in hourly intervals which, ultimately,
were aggregated to daily totals from 0 to 0 CST for each day used in the study. Forty-six days for the year
of 2014 were analyzed for a total of 1,104 hours for each radar which converts to, approximately, 33,000
radar scans in all. The days were chosen based on availability of data from the National Climate Data
Center's (NCDC) Hierarchal Data Storage System (HDSS) for all three radars, in addition to error-free
performance notes from each of the gauges used. The dates analyzed were split near evenly between
warm (May – October) and cool (November – April), therefore encompassing an overall performance of
each of the radars throughout the year with no preferential bias towards rain or snow. Additionally, days
were distributed evenly during the summer between convective and stratiform events with a threshold of
38 dBZ (Gamache and Houze, 1982).
Observed precipitation data were collected using Campbell Scientific TE525 tipping buckets
located at each of the locations for the study (Table 1). The precipitation gauges have a 15.4-cm orifice
which funnels to a fulcrum which registers 0.01 mm of rainfall per tip. The performance of each gauge is



maximized between 0 and 50°C, for which each day of the study's temperature did not exceed. Accuracy
in gauge measurements range between -1 to 1%, -3 to 0%, and -5 to 0% for precipitation up to 25.4 mm h$^{-1}$
$^{1}$, 25.4 to 50.8 mm h$^{-1}$, and 50.8 to 76.2 mm h$^{-1}$, respectively, which are, primarily, associated with local
random errors and errors in tip-counting schemes (Kitchen and Blackall, 1992; Habib et al., 2001). Each
tipping bucket is located, approximately, 1 m above the ground in areas clear of buildings and properly
maintained vegetation height to mitigate turbulence effects (Habib et al., 1999). These errors were
assumed negligible and, therefore, allowed for the gauges to be representative of the true rainfall rate.
**2.3 Radar data and radar-rainfall algorithms**
Next Generation Radar (NEXRAD) level-II data were retrieved from the NCDC's HDSS. Files
were analyzed using the Weather Decision Support System – Integrated Information (WDSS-II) program
(Lakshmanan et al., 2007) to assess reflectivity (Z) in addition to dual-polarized radar variables including
differential reflectivity (ZDR) and specific differential phase shift (KDP). Three other variables were also
generated based on a KDP-based smoothing field (Ryzhkov et al., 2003) for reflectivity, differential
reflectivity, and specific differential phase: DSMZ, DZDR, and DKDP, respectively. A rain rate echo
classification variable (RREC) was also computed, which chooses whether an R(Z), R(KDP), R(Z,ZDR),
or R(ZDR, KDP) algorithm is implemented in estimating rain rates based on the radar fields of Z, ZDR,
and KDP (Kessinger et al., 2003).
All seven variables (Z, ZDR, KDP, DSMZ, DZDR, DKDP, and RREC) were converted from
their native polar grid to 256 x 256 1-km Cartesian grids, where the lowest radar elevation scans (0.5°)
were used to mitigate uncalculated effects from evaporation and wind drift. An average of 5-minute scans
were used for each of the variables, which were aggregated to hourly totals to be compared to the hourly
tipping-bucket accumulations. The latitude and longitude of each of the 15 gauges were matched with the
radar pixel that corresponds to the Cartesian grid such that each quantitative value of the seven radar
variables were able to be extracted and used in rain rate calculations. Post-processing rain-rate





calculations were conducted using the equations presented by Ryzhkov et al. (2005) (Table 2), which
were gathered from multiple studies using disdrometers to derive a relationship between reflectivity,
differential reflectivity, and specific differential phase (Bringi and Chandrasekar, 2001; Brandes et al.,
2002; Illingworth and Blackman, 2002; Ryzhkov et al., 2003). Standard R(Z) algorithms were also
included to test whether the addition of dual-polarized technology to rainfall estimates produced
improvement.

With the use of both Z, ZDR, KDP, and DSMZ, DZDR, and DKDP fields produced by WDSS-II,

the number of algorithms tested was 55. This includes the three standard single-polarized algorithms
(stratiform, convective, and tropical) which were calculated using reflectivity R(Z), and then calculated as
R(DSMZ), while algorithms 1-6 (R(KDP)) were also calculated as R(DKDP). Algorithms 7-11 (R(Z,
ZDR)) were additionally calculated as R(Z, DZDR), R(DSMZ, ZDR), and R(DSMZ, DZDR), while the
same four combinations of non- and KDP-smoothed fields were applied to the R(KDP, ZDR) algorithms

(12-15).

**2.4 Statistical analyses**

To test the performance of each algorithm, several statistical analyses were calculated. The

average difference (Bias) was calculated as
$$Bias = \frac{\sum (R_i - G_i)}{N}$$    (1)
where $R_i$ is each hourly aggregated radar estimated rainfall amount calculated from one of the 55
algorithms, $G_i$ is the hourly aggregated gauge (observed) measurement, and $N$ is the total number of
observations which, for this study, was 1,104 hours. A second statistical parameter, the normalized mean
bias (NMB), was calculated as



$$NMB = \frac{1}{N} \frac{\sum (R_i - G_i)}{\sum G_i}$$ (2)
The normalized mean bias is included in the analyses due to the fact that overestimations (i.e., radar
estimates larger than gauge measurements) and underestimations (i.e., radar estimates smaller than gauge
measurements) are treated proportionately. This is directly analogous to choosing the mean absolute error
(MAE) opposed to the standard deviation as the MAE does not penalize smaller or larger errors,
obscuring the overall results (Chai and Draxler, 2014). Bias measurements (Bias and NMB) were
calculated to determine whether radar derived rain rates were over- or under-estimated in comparison to
the gauges. However, to calculate the overall magnitude of error associated with the performance of the
radars, the absolute values of (1) and (2) were performed to yield the mean absolute error (MAE), and
normalized standard error (NSE), respectively.
Several other meteorological parameters were calculated, including probability of detection
(PoD) which was calculated as
$$PoD = \frac{\sum |R_i \bullet G_i > 0 \ \& \ R_i > 0|}{\sum |G_i|}$$ (3)
where the bullet ($\bullet$) indicates "if", to determine how accurate the radars were at correctly detecting
precipitation. The probability of detection values range between 0.0 (radar did not detect any precipitation
correctly) and 1.0 (radar detected the occurrence of all precipitation 100% correctly). The probability of
false detection takes into account the amount of precipitation the radars incorrectly estimated when the
gauges recorded zero values, and was calculated as
$$PoFD = \frac{\sum R_i \bullet (G_i = 0 \ \& \ R_i > 0)}{\sum G_i}$$ (4)
Conversely, the missed precipitation amount (MPA) is the opposite of the PoFD, such that



$$MPA = \sum R_i \bullet (G_i > 0 \ \& \ R_i = 0) \qquad (5)$$

Equations 3, 4, and 5 are scaled by the amount of precipitation measured by the gauges. The total amount
of rainfall missed and falsely detected (i.e., numerator) of (3), (4), and (5) were also quantified and
reported.

**3    Results and discussion**
**3.1 Individual radar performance: All data**
To test the overall performance of each radar, it was necessary to determine the overall best
algorithm for each statistical measure. Furthermore, the algorithm that performed the best and worst for
each gauge and for each radar was assessed.

**3.1.1 KEAX**
The overall bias showed that there was a positive bias, peaking near 5.5 mm hr$^{-1}$ at the second
gauge for KEAX, approximately 115 km from the radar for both the best and worst performing
algorithms (Figure 2). This could correspond to a bright-band signature which caused overestimation in
precipitation from the algorithms. The overall worst algorithm, equation 13, an R(ZDR,KDP)
relationship, revealed a decreasing trend in bias as the distance from the radar increased. This could be
due, at least in part, to the algorithm's utilization of KDP which performs poorly in frozen precipitation
(Zrnic and Ryzhkov, 1996), causing the underestimation. Conversely, the algorithm with the lowest bias
was an R(Z,ZDR) algorithm (equation 11). There was a maximum in the bias calculations while utilizing
equation 11 near 120 km, similar to equation 13, however, there was a more pronounced minimum in the
data near 150 km. Furthermore, it appears the data oscillates around a bias value of 0 mm hr$^{-1}$ when using





equation 13. This could be due to ZDR's capability to respond to precipitation shape (Kumjian 2013a, b),
which helps to scale the reflectivity portion of the rainfall estimation algorithm to a more accurate value.
The normalized mean bias (NMB) reveals the same trend in values for bias but with a decrease in
magnitude. It is important to note, however, that the algorithms that tend to perform the worst (e.g.,
algorithms containing KDP) result in anomalous range responses which would be due, at least in part, to a
stronger response to precipitation type.

The absolute bias and normalized standard error (NSE) shows the same maxima in the data at the

second gauge (Brunswick) that was present in the bias data. However, a second maxima is located at the
fifth gauge at, approximately, 150 km (Linneus), which could be a second bright-band present in the
summer data, whereas the first maxima is a bright-band in the winter data. There was also a more
pronounced minimum in the NSE results at the fourth gauge, indicating the effects of stratiform as
opposed to convective precipitation.

The probability of detection (PoD) results show a large difference in algorithm choice for

correctly detecting precipitation. The KDP-smoothed R(Z) convective algorithm, R(DSMZ) convective,
performed the best in terms of correctly detecting precipitation, whereas algorithm 1 (KDP1) performed
the worst, despite its advantages at large ranges (Zrnic and Ryzhkov, 1996). The increased PoD at the
second gauge indicates the definite presence of a bright-band, while the low PoD at, approximately 150
km, indicates overshooting of the beam. This is further aided by the MPA results, as about 225 mm of
precipitation was missed by the radar at 150 km, whereas only 100 mm of precipitation was missed by the
radar at the second gauge at 120 km. Although equation 11, an R(Z,ZDR) algorithm was superior in terms
of the bias, the same algorithm with a KDP-smoothed reflectivity value, R(DSMZ,ZDR) revealed the
overall least amount of falsely missed precipitation. However, the summation of the amount of
precipitation falsely detected (PoFD) by KEAX showed a larger source of error than the MPA in terms of



magnitude. For example, at the second (fifth) gauge, only 100 (225) mm of precipitation was missed by
the radar, but over 700 (725) mm of precipitation was incorrectly estimated by the radar.

**3.1.2 KLSX**
Unlike the KEAX data, the gauges used for analyses for the KLSX radar span between 90 – 150
km. Furthermore, 5 out of the 8 gauges were located within 10 km of range from one-another, near 140
km from the radar, limiting the data available for analyses between 100 and 140 km (Figure 3).
The bias and NMB show a relatively modest peak in values near the second gauge of 5 mm hr$^{-1}$,
which decreases to approximately 3.6 mm hr$^{-1}$ at the third gauge, 120 km from the radar. The worst
performing algorithm, equation 13, was the same R(ZDR,KDP) relation as the worst KEAX bias and
NMB data. Additionally, the overall trend of decreasing bias and NMB as distance from the radar
increases was noted, presumably due to overshooting effects similar to the KEAX data. Furthermore, the
overall negative bias displayed by the best-performing algorithm, equation 11, was similar to the KEAX
data as well.
The double maxima in the absolute bias graph are present as with the KEAX data, but are not as
pronounced. Additionally, the overall minima in the absolute bias for both KEAX and KLSX are at,
approximately, 125 km from the radar. However, the relative distance from the radars are the same, where
the two maxima for KEAX were at 115 and 150 km, while the maxima were at, approximately, 100 and
140 km. The overall best and worst performing algorithms for the absolute bias and NSE were equations
11 and 13, the R(Z,ZDR) and R(ZDR,KDP) algorithms, respectively.
One of the main differences between the KLSX and KEAX data was the decreased probability of
detection at 120 km for KLSX, while there was an increased probability of detection for KEAX. In
general, the PoD values were worse for KLSX when compared to KEAX. There was also a trend of



increasing PoD values as distance from the St. Louis radar increased and, at one point near 140 km, the
best algorithm, R(DSMZ) convective and the worst algorithm, KDP1, were not significantly different
(10% difference in detection). Additionally, the maxima in the PoD while utilizing KDP1 corresponds to
a minima in the R(DSMZ) detection percentage, which is well correlated by the similarly valued MPA
results.

Another difference between the KEAX and KLSX data was the overall decrease in the PoFD as

distance from the radar increased. Because of this, the maxima in the amount of falsely identified
precipitation is only 100 km from the radar, which may be effects from bright-banding. Furthermore, this
resulted in the overall error in precipitation for algorithm 13 to be in excess of 1,500 mm, while algorithm
11 did not exceed 500 mm for the 1,104-hour dataset for KLSX.

**3.1.3 KSGF**

Although the KLSX and KEAX data strongly suggests bright-banding signatures near

approximately 100 km and 150 km from the radar, the KSGF results reveal an overall increase of error
with range (Figure 4). One of the main reasons for this could be due to the fact that the gauge furthest
from any radar analyzed is Cook Station, 185 km from KSGF, which is the range where Ryzhkov et al.
(2003, 2005) reported significant fallout in radar performance in rainfall estimation.

Overall, the absolute bias values for KLSX, KEAX, and KSGF were within $\pm$ 2 from 6 mm hr$^{-1}$

for the worst performing algorithm, equation 13. However, the radar at Springfield, MO revealed the
maximum absolute bias was the furthest gauge at, approximately, 185 km (Cook Station). Although a
slight bright-band effect is evident at the second gauge, 100 km from KSGF, the first bright-band is not as
evident when compared to the KEAX and KLSX data. However, the overshooting of the beam is more
pronounced between 140- and 160-km from KSGF. For example, there is a sharp decrease in the
probability of detection within this range, correlating with a decrease in the bias and NMB. Furthermore,





there is an increase in the magnitude of the FAR, indicating a large portion of precipitation was no
captured by the radar beam.

**3.2 Individual radar performance: Seasonal data**
In order to achieve a better understanding of the minimum and maximum values portrayed by the
data, all of the radar scans and gauge data were divided into summer (May – October) and winter
(November – April) months based on the average climatology of Missouri. This resulted in 652 hours of
data for summer, and 452 hours for winter (59 and 41% of the entire data, respectively). Because of this,
the overall error is more weighted towards the summer data than the winter data.
The Kansas City bias and absolute bias summer data (Figure 5) shows a similarity to the overall
data (Figure 2) in terms of both trend and magnitude. Also, the best performing algorithm for the
probability of detection (equation 11) was the same for the summer and overall data. However, the R(Z)
Tropical algorithm showed the least reliability in correctly detecting precipitation for the summer,
resulting in a more pronounced decrease in the PoD percentage overall. For the NMB and NSE data, the
same algorithms that performed best and worse for the overall data (equations 11 and 13) were the best
and worst for summer, respectively, and showed similar magnitudes and trends. Conversely, the winter
data (Figure 6) showed a pronounced overestimation in the NMB and NSE at the third gauge (125 km)
from the radar, with values exceeding 30 mm hr$^{-1}$ compared to values below 6 mm hr$^{-1}$ for the combined
seasonal data. This could be due, at least in part, to the large amount of precipitation overestimated by the
radar relative to the total amount of precipitation. For example, winter precipitation amounts are
significantly lower than convective summertime amounts and, thus, result in a small denominator in (2),
leading to an increase in bias. These trends in the KEAX higher NMB and NSE values can be observed
for the KLSX and KSGF data as well (Figures 7 and 8, respectively). However, the magnitudes of NMB
and NSE were smaller for KSGF in comparison to KLSX and KEAX.



Summing the amount of precipitation not recorded by the radar but recorded by the gauge (MPA)
showed similar results when compared between summer and the overall data, but also revealed little
contribution of the overall amount from the winter data. Additionally, the best and worst algorithms for
the MPA (equations 10 and 14, respectively) were not significantly different ($p = 0.05$). Furthermore, the
relatively small contribution from the winter data to the amount of precipitation not estimated by the
gauge but estimated by the radar (30 and 40 mm for KLSX and KSGF, respectively) was similar to the
KEAX data. The contributions of winter MPA to the overall MPA for all three radars were,
approximately, 20%. Conversely, the total amount of precipitation recorded by the radar but not recorded
by the gauge (PoFD) showed a relatively large portion from the winter data as opposed to the summer
data, with the noticeable exception of the bright-banding effects at the second and fifth gauges (120 and
150 km, respectively) for KEAX. Overall, the winter contribution to the overall PoFD was about 50%.
Overall, the summation of all errors from the radar, including MPA, PoFD, and the absolute bias
reveals that, approximately, 20-30% of the error was due to the winter data while comprising 41% of the
entire dataset for all three radars. Conversely, the bulk of the error (80%) was due to the 59% total
summer results, primarily due to the overall larger magnitudes in rainfall from convective storms. This is
further exemplified via Figure 9, showing a scatterplot of all gauge versus radar comparisons. With the
exception of a few data points for KEAX, seldom does the winter radar estimated precipitation exceed 10
mm hr$^{-1}$, while no gauge recorded precipitation exceeded 10 mm hr$^{-1}$. It is interesting to note that, with the
exception of the KLSX data, the winter correlation coefficient values exceed the summer. This could be
due, at least in part, to local random errors (Ciach and Krajewski, 1999a) and the excessive (i.e.,
convective) rainfall that the tipping buckets are unable to accurately measure (Ciach and Krajewski,
1999b; Ciach 2002). Furthermore, because the magnitude of precipitation in the winter is less than the
summer, smaller variance and absolute error values are common, causing the correlation coefficient
values to be larger than the frequent summertime showers where precipitation values can range from 0
mm hr$^{-1}$ to, in extreme cases, 100 mm hr$^{-1}$.




### 3.3 Radar performance: Hits only

From the results presented thus far, the majority of the error has resulted from either the PoFD or
MPA. Therefore, an analysis into how accurate each algorithm was in comparison to a one-to-one ratio
for a correct hit (i.e., gauge and radar recorded precipitation) is presented. This will, in turn, determine
whether algorithm 11, an R(Z,ZDR) equation, is still most accurate and whether algorithm 13, an
R(ZDR,KDP) equation, is least accurate.
From the 55 algorithms possible, the first gauge from each of the three radars (Greenridge,
Williamsburg, and Lamar for KEAX, KLSX, and KSGF, respectively), all within 100 km from the radar,
showed that either an R(Z) or R(DSMZ) convective algorithm was most accurate with correlation
coefficient values around 0.70 (Figures 10-12). Additionally, the second gauge from KEAX (Brunswick)
also revealed that an R(Z) convective $R^2$ value was superior to all other algorithms. For the intermediate
gauges from each radar, the rain rate echo classification (RREC) algorithm had the highest correlation
coefficient value. For example, for KEAX, St. Joseph (115 km) and Versailles (129 km) had some of the
highest $R^2$ values of 0.62 and 0.88, respectively. For KLSX, the fourth gauge (Bradford, at 135 km from
the radar), the RREC correlation coefficient value was 0.55. Beyond, approximately, 140 km from the
radar, the KDP3 equation was superior. In fact, the furthest two gauges from each radar showed KDP3 $R^2$
values exceeding 0.40. This could be due, at least in part, to the fact that the specific differential phase
does not degrade quality with range, resulting in more accurate results at larger distances (Zrnic and
Ryzhkov, 1999; Ryzhkov et al., 2003).
For the vast majority of scenarios, DZDRDKDP2 or the R(Z) Tropical algorithms were the worst
performing equations. Because the R(Z) Tropical equation was designed for maritime precipitation while
this study was conducted in the Midwest, it was not surprising that it was one of the poorest performing
algorithms. From the scatterplots of gauge versus radar precipitation (Figures 10-12), when the R(Z)





Tropical equation was the worst correlation correlation-valued algorithm, there was, generally,
underestimation of precipitation estimated by the radar. Conversely, for the DZDRDKDP algorithm,
overestimation of radar estimated precipitation was observed. This could be due to over-smoothing of the
ZDR and KDP fields, causing overestimation in rain estimates (Simpson et al., 2016).



**4     Conclusions**

Dual-polarization technology was implemented to the National Weather Service Next Generation

Radar network in the Spring of 2012 to, primarily, improve precipitation estimation and hydrometeor
classification. Since this time, a number of studies have been conducted to determine whether this
upgrade has improved radar performance in a meteorological, and hydrometeorological sense. Many
studies have observed an improvement of radar-based precipitation estimation compared to terrestrial-
based precipitation monitors (e.g., Ryzhkov et al., 2003, 2005; Simpson et al., 2016), while other studies
show ambiguity between whether there is improvement (e.g., Gourley et al., 2010; Cunha et al., 2015).
This study observed over 1,100 hours of precipitation data with three separate radars in Missouri using 55
algorithms including the three conventional R(Z) radar rain-rate estimation algorithms (stratiform,
convective, and tropical) along with a myriad of R(KDP), R(Z,ZDR), and R(ZDR,KDP) algorithms
which can be found in Ryzhkov et al. (2005). Additionally, a KDP-smoothing field of reflectivity,
differential reflectivity, and the specific differential phase shift (DSMZ, DZDR, and DKDP, respectively)
were measured and used for analyses. Unlike previous studies, the current work emphasizes the amount
of precipitation correctly and incorrectly estimated by the radar in comparison to the terrestrial based
precipitation gauges through measurements of the probability of detection, probability of false detection,
and missed precipitation amount.



For all three radars, Kansas City, St. Louis, and Springfield, MO (KEAX, KLSX, and KSGF,

respectively), the vast majority of precipitation error (over 60%) was contributed by the amount of

precipitation falsely detection by the radar (up to 725 mm), while 20% was due to the radar missing the

precipitation (up to 225 mm) for KEAX. Similar magnitudes of error were reported for KLSX and KSGF,

with an overall error in precipitation for each radar ranging between 250 mm for the best performing of

the 55 algorithms, equation 11 (an R(Z,ZDR) algorithm), and up to 2000 mm for the worst performing

algorithms, R(ZDR,KDP) equation 13.

Radar performance in different seasons has been shown to be significantly different, therefore,

the data was divided into summer (May – October) and winter (November – April) months resulting in

652 hours for summer, and 452 hours for winter (59 and 41% of the entire data, respectively).  Despite the

winter data contributing less than the summertime data, it accounted for 20% of the overall MPA, and

40% to the overall PoFD. The best and worst performing algorithms were the same for the summer and

winter data as the overall data, R(Z,ZDR) equation 11 and R(ZDR,KDP) equation 13, respectively.

The overall data was further subdivided into correct radar hits (radar correctly estimated

precipitation to be present while the terrestrial based gauge recorded precipitation) for the 1,100-hour

dataset. It was found that within 100 km from each of the three radars, the R(Z) or R(DSMZ) convective

algorithm revealed the best correlation coefficient values of, approximately, 0.70. Further from the radar,

beyond 135 km, RKDP3 generally performed the best due to the algorithms non-degrading capabilities

and immunity to beam blockage, whereas at intermediate distances (between 100 and 135 km from the

radars), the rain rate echo classification algorithm performed the best. Overall, the worst performing

equations were either the R(Z) tropical, or DZDRDKDP2.

These results help our understanding in the possibilities for hydrometeorological studies.

Although a mixture of R(Z) convective and R(KDP) algorithms performed the best when precipitation

was correctly estimated by the radar, nearly 50% of the 1,100 hours analyzed for the study consisted of



either falsely estimated precipitation by the radar, or missed by the radar. Furthermore, these errors
accumulate between 500 to 2,000 mm of precipitation depending on the algorithms chosen. Because of
this, a significant source of error and uncertainty must be overcome before radar data can be fully
implemented into hydrologic models, especially on a continuous, operational basis.

**Author Contribution.** N. Fox designed the experiment and provided feedback while M. Simpson carried
out the calculations and wrote the manuscript.
**Acknowledgements.** This material is based upon work supported by the National Science Foundation
under Award Number IIA-1355406. Any opinions, findings, and conclusions or recommendations
expressed in this material are those of the authors and do not necessarily reflect the views of the National
Science Foundation.

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















Table 1. Terrestrial-based precipitation gauge locations used for the study in addition to the National
Weather Service Radars Springfield, MO (KSGF), Kansas City, MO (KEAX), and St. Louis, MO
(KLSX) used in conjunction with each gauge.

| Gauge Location | Latitude (°N) | Longitude (°W) | Radar(s) Used |
|---|---|---|---|
| Bradford | 38.897236 | -92.218070 | KLSX, KEAX |
| Brunswick | 39.412667 | -93.196500 | KEAX |
| Capen Park | 38.929237 | -92.321297 | KLSX, KEAX |
| Cook Station | 37.797945 | -91.429645 | KLSX, KSGF |
| Green Ridge | 38.621147 | -93.416652 | KEAX, KSGF |
| Jefferson Farm | 38.906992 | -92.269976 | KLSX, KEAX |
| Lamar | 37.493366 | -94.318185 | KSGF |





| | | | |
|---|---|---|---|
| Linneus | 39.856919 | -93.149726 | KEAX |
| Monroe City | 39.635314 | -91.725370 | KLSX |
| Mountain Grove | 37.153865 | -92.268831 | KSGF |
| Sanborn Field | 38.942301 | -92.320395 | KLSX, KEAX |
| St. Joseph | 39.757821 | -94.794567 | KEAX |
| Vandalia | 39.302300 | -91.513000 | KLSX |
| Versailles | 38.434700 | -92.853733 | KEAX, KSGF |
| Williamsburg | 38.907350 | -91.734210 | KLSX |






Table 2. List of single- and dual-polarimetric algorithms used for radar rainfall estimates.

| $R(Z) = aZ^b$ | | | |
|---|---|---|---|
| Precipitation type | a | b | c |
| Stratiform | 200 | 1.6 | - |
| Convective | 300 | 1.4 | - |
| Tropical | 250 | 1.2 | - |
| $R(KDP) = a \mid KDP \mid^b sign(KDP)$ | | | |
| Algorithm number | | | |
| 1 | 50.7 | 0.85 | - |





| | | | |
|---|---|---|---|
| 2 | 54.3 | 0.81 | - |
| 3 | 51.6 | 0.71 | - |
| 4 | 44.0 | 0.82 | - |
| 5 | 50.3 | 0.81 | - |
| 6 | 47.3 | 0.79 | - |

$R(Z, ZDR) = aZ^b ZDR^c$

| Algorithm number | | | |
|---|---|---|---|
| 7 | $6.70 \times 10^{-3}$ | 0.927 | -3.43 |
| 8 | $7.46 \times 10^{-3}$ | 0.945 | -4.76 |
| 9 | $1.42 \times 10^{-2}$ | 0.770 | -1.67 |
| 10 | $1.59 \times 10^{-2}$ | 0.737 | -1.03 |
| 11 | $1.44 \times 10^{-2}$ | 0.761 | -1.51 |

$R(ZDR, KDP) = a \, | \, KDP \, |^b \, ZDR^c \, sign(KDP)$

| **Algorithm number** | | | |
|---|---|---|---|
| 12 | 90.8 | 0.930 | -1.69 |
| 13 | 136 | 0.968 | -2.86 |
| 14 | 52.9 | 0.852 | -0.53 |
| 15 | 63.3 | 0.851 | -0.72 |




















**Figures**



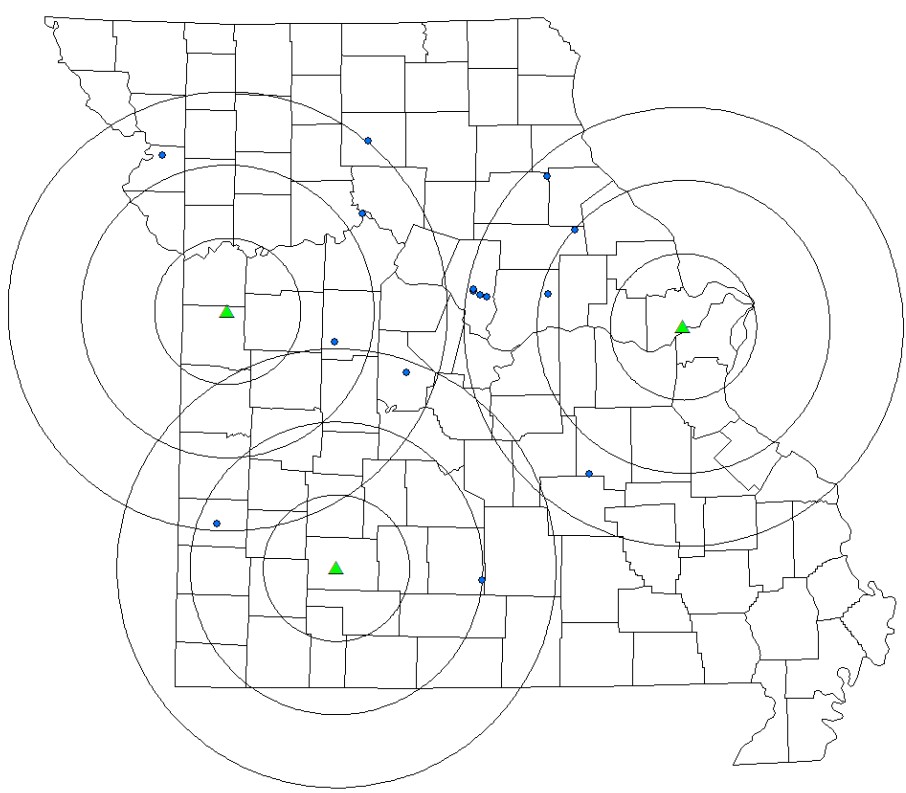


Figure 1. Study location (Missouri) with St. Louis (KLSX), Kansas City (KEAX), and Springfield
(KSGF), MO radars (triangles) overlaid with 50-, 100-, and 150-km range rings in addition to the 15
terrestrial-based precipitation gauges utilized as ground-truthed data.





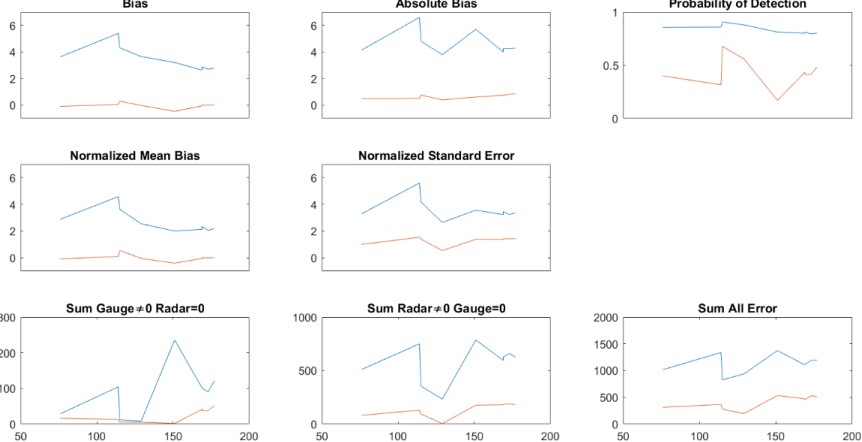


Figure 2. Overall statistical analyses for the nine gauges used for Kansas City, MO. The blue line
represents the weakest performing rain rate estimation algorithm, while the red line represents the overall
best performing algorithm for all graphs, with the exception of the probability of detection. All units are
in mm hr$^{-1}$ with the exclusion of the probability of detection (unitless).










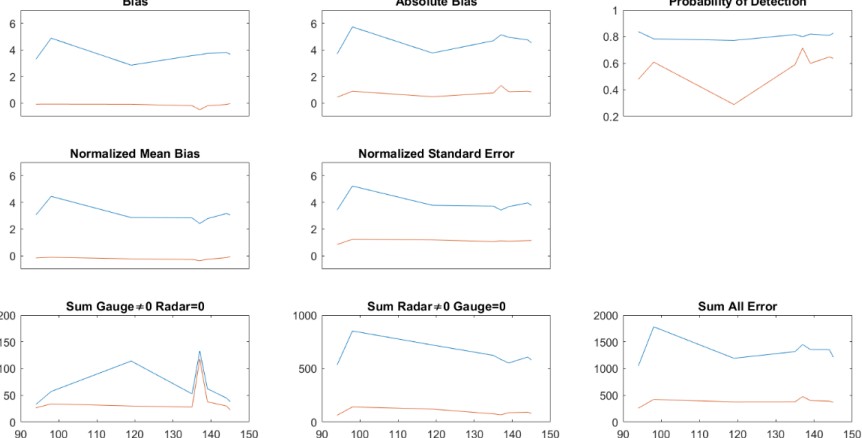


Figure 3. Overall statistical analyses for the nine gauges used for St. Louis, MO. The blue line represents
the weakest performing rain rate estimation algorithm, while the red line represents the overall best
performing algorithm for all graphs, with the exception of the probability of detection. All units are in
mm hr$^{-1}$ with the exclusion of the probability of detection (unitless).








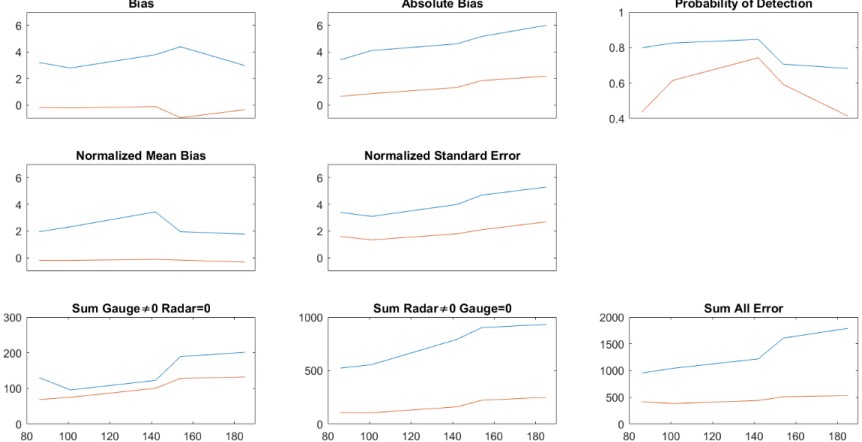

Figure 4. Overall statistical analyses for the nine gauges used for Springfield, MO. The blue line
represents the weakest performing rain rate estimation algorithm, while the red line represents the overall
best performing algorithm for all graphs, with the exception of the probability of detection. All units are
in mm hr$^{-1}$ with the exclusion of the probability of detection (unitless).








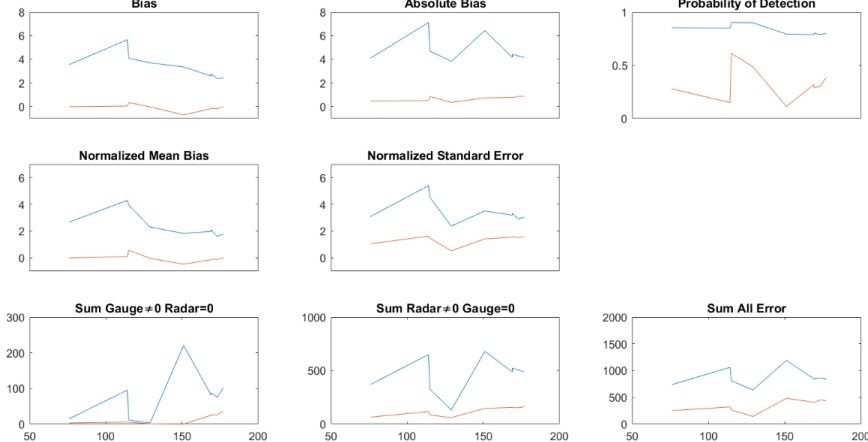


Figure 5. Statistical analyses for the nine gauges used for Kansas City, MO for warm season data, only.
The blue line represents the weakest performing rain rate estimation algorithm, while the red line
represents the overall best performing algorithm for all graphs, with the exception of the probability of
detection. All units are in mm hr$^{-1}$ with the exclusion of the probability of detection (unitless).







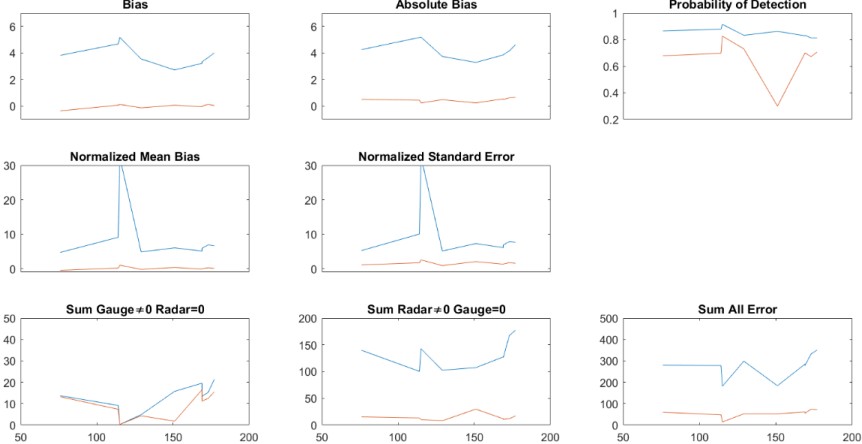


Figure 6. Statistical analyses for the nine gauges used for Kansas City, MO for cool season analyses, only.
The blue line represents the weakest performing rain rate estimation algorithm, while the red line
represents the overall best performing algorithm for all graphs, with the exception of the probability of
detection. All units are in mm hr$^{-1}$ with the exclusion of the probability of detection (unitless).

590       .





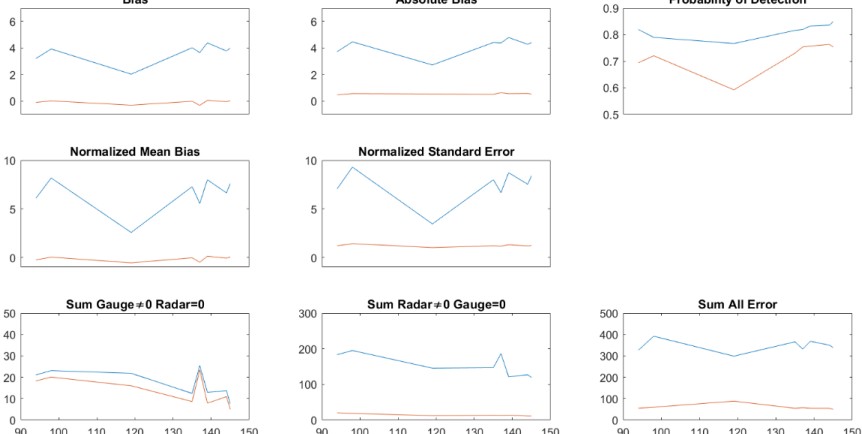


Figure 7. statistical analyses for the nine gauges used for St. Louis, MO for cool season analyses,
only. The blue line represents the weakest performing rain rate estimation algorithm, while the red
line represents the overall best performing algorithm for all graphs, with the exception of the
probability of detection. All units are in mm hr$^{-1}$ with the exclusion of the probability of detection
(unitless).




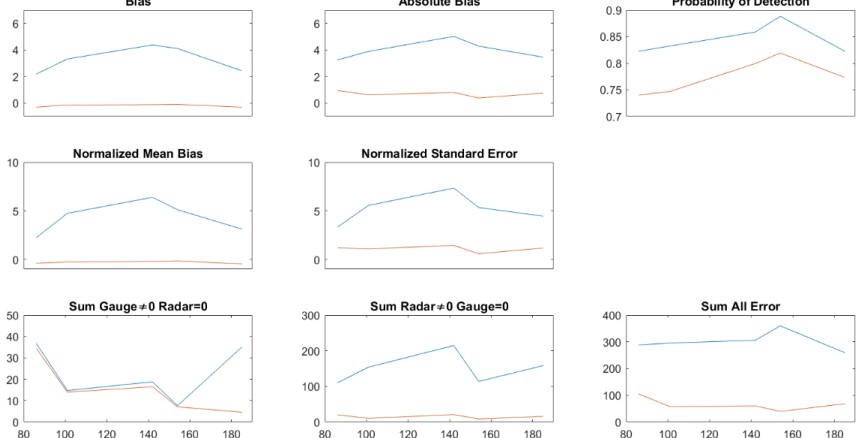


Figure 8. Statistical analyses for the nine gauges used for Springfield, MO for cool season analyses,

only. The blue line represents the weakest performing rain rate estimation algorithm, while the red

line represents the overall best performing algorithm for all graphs, with the exception of the

probability of detection. All units are in mm hr$^{-1}$ with the exclusion of the probability of detection

(unitless).








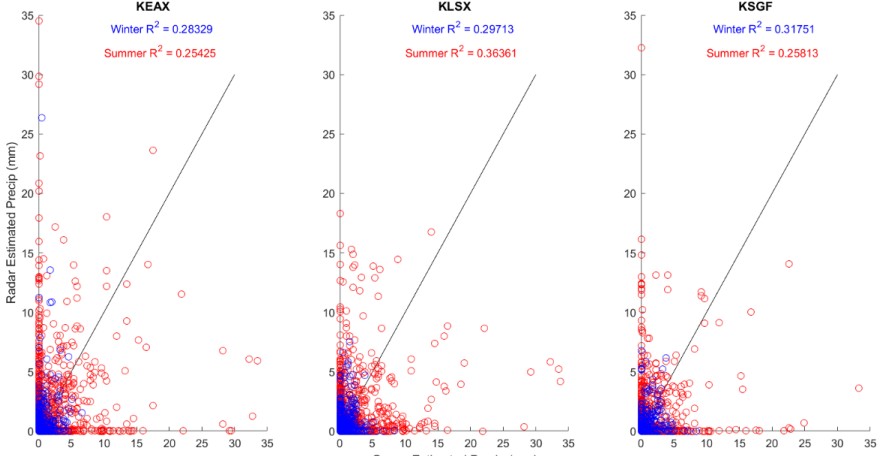


Figure 9. Scatterplot of gauge estimation precipitation versus radar estimated precipitation with their

respective correlation coefficient values for warm and cool seasons.









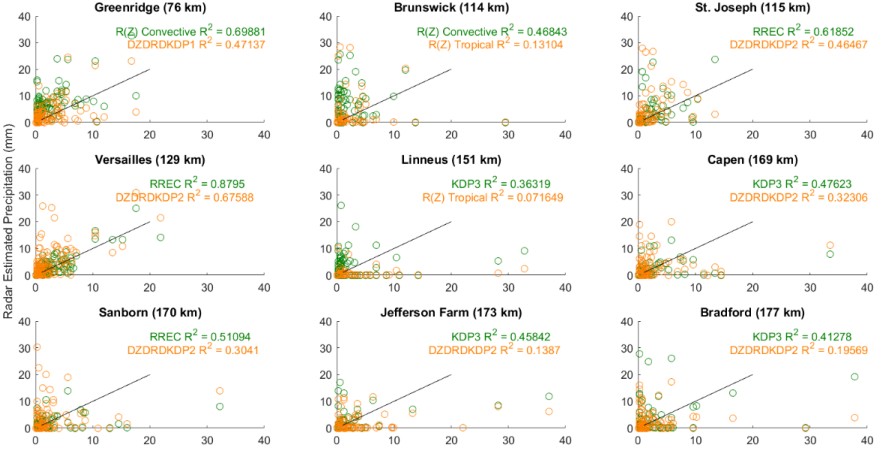


Figure 10. Scatterplots of the best (green) and worst (orange) performing radar rain rate estimation

algorithms at each terrestrial based gauge location. Distance from the Kansas City (KEAX) radar is

labeled in parenthesis next to the gauge name.








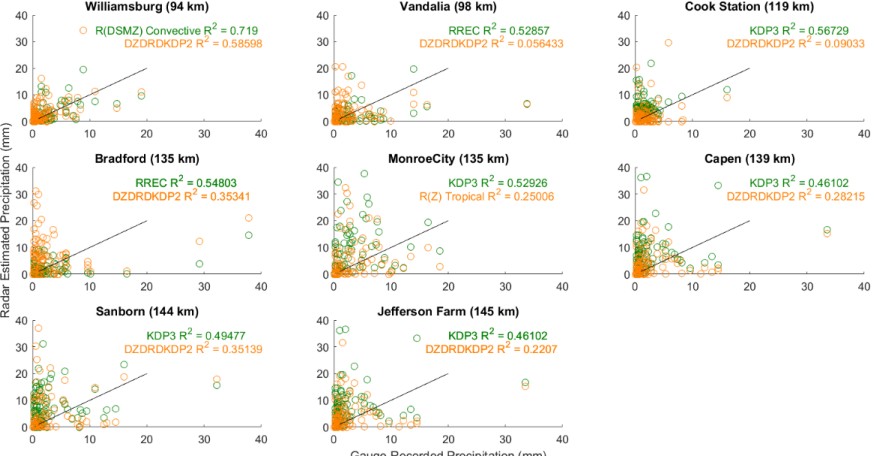


Figure 11. Scatterplots of the best (green) and worst (orange) performing radar rain rate estimation

algorithms at each terrestrial based gauge location. Distance from the St. Louis (KLSX) radar is

labeled in parenthesis next to the gauge name.











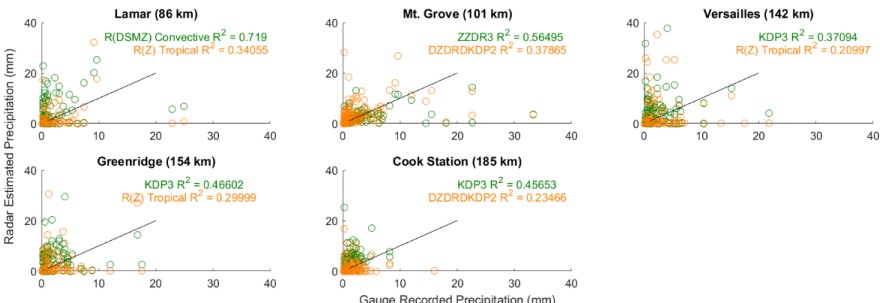


Figure 12. Scatterplots of the best (green) and worst (orange) performing radar rain rate estimation
algorithms at each terrestrial based gauge location. Distance from the Springfield (KSGF) radar is
labeled in parenthesis next to the gauge name.