# Peer review of "DUAL-POLARIZED QUANTITATIVE PRECIPITATION ESTIMATION AS A FUNCTION OF RANGE"

_Hydrology and Earth System Sciences, 2017_

## Referee Comment (RC1) · E. Goudenhoofdt (Referee) · 4 Aug 2017

**Comments to "Multi-radar performance in the Midwestern United States at large ranges"**

Edouard Goudenhoofdt

August 4, 2017

**1 General comments**

This paper analyses the performance of several rainfall estimation algorithms from 3 polarimetric radars in the midwestern united states. Such verification studies are not particularly original but constitute a good contribution to the field. Unfortunately the limited amount of data used in the present study reduces its value.

The scientific quality of the paper is poor due to a limited literature review, questionable methods and results which are not sufficiently discussed and lack of significant conclusions.

The presentation is neither clear nor concise and some inconsistencies appear in the results. The text is particularly difficult to read due to inappropriate use of the language.

Despite those flaws I think interesting results on the performance of dual-pol radars can be shared with the community. Therefore major comments are given below to help the authors to reach an acceptable level of quality to consider publication.

**2 Major comments**

**2.1 Does the paper address relevant scientific questions within the scope of HESS?**

The study of the performance of radar-based precipitation estimation is clearly in the scope of HESS. However the motivation for this study should be better explained in the introduction. What are the possible applications ?

The specific objectives of the paper are detailed at the end of the introduction. While studying the effect of range is interesting, do you think it justifies the whole paper? Is the added value of dual-pol variables not the main focus of the paper? What do you exactly mean by testing the best radar algorithm? On average? Did you consider comparing the performance of the 3 radars since calibration of the dual-pol measurements is crucial. The rationale for separating the data in warm and cool seasons is not sufficiently explained. What about grouping the data by month or weather regimes?

**2.2 Does the paper present novel concepts, ideas, tools, or data?**

There is nothing particularly new in the paper but long-term verifications of precipitation estimates from dual-pol radars are welcome.

The authors present the paper as an extension of a previous work with more data but it is limited to the year 2014. Since they are easily available in the US, why not using the radar data from 2013-2016? Using 4 years of data would decrease the statistical biases and be more representative of the weather in Missouri. I understand that only the hours when all radar and gauge data are available are taken but what explains the very low availability (13 %) and how many of the 1100 hours are dry episodes? This results in a limited dataset subject to statistical biases.

**2.3 Are substantial conclusions reached?**

In general the limitations in the data and presented results do not allow the derivation of substantial conclusions.

The paper concludes that errors come mainly from mismatch with rain-gauges either in winter or summer. For "all" data the R(Z,ZDR) is the best and R(ZDR, KDP) the worst while for "hits" data the best is R(Z) up to 100 km, R(KDP) from 135 km and the algorithm based on the type of precipitation in between.

Important results from the paper are not reported in the conclusions like the bias and error in function of the range. The strong impact of false and missed detection is not sufficiently discussed. It is often unclear if the results are valid for all algorithms or a selection of them. The fact that the best algorithm is different if you consider "all" or "hits only" data is not sufficiently discussed. There is not much on the added value of dual-pol compared to single-pol. Most results (not only in the conclusions) are presented in a qualitative way (best, worst, increase) while information on the magnitude of the (relative) performance is missing.

**2.4 Are the scientific methods and assumptions valid and clearly outlined?**

Some parts of the methodology are questionable.

The rain gauge network is small and not homogeneous and the gauges tend to be far from the radars. This make the statistical analysis difficult especially the study of range effects. These limitations are not properly discussed through the text.

The information on the quality of the radar and raingauge measurements used in this study is not sufficient and too general. Did the authors perform any complementary control on the data? The quality of the datasets is important to obtain robust results and avoid artefacts.

The presentation of the rainfall estimation algorithms is unclear. Do you accumulate the dual-pol variables over one hour ? What is the rationale of using 11 different algorithms (looking at the table some algorithms have similar parameters)? Is it necessary to make all the combination of smoothed and original fields? The "RREC" algorithm is not well described. Is it fair to use this advanced algorithm?

You focus only on the rainfall estimation from radar variables. However processing weather radar data includes important steps like clutter mitigation and correction for the height of the measurements. Is this not available in WDSS? Did you consider using operational rainfall estimation products as reference? Did you consider compositing your 3 radars to obtain a better overall performance? Did you consider using multi-sensor products?

The verification methodology is questionable :

- Why not using only the normalised statistics ?

- Did you consider conditional statistics (e.g. only values above $1\,\mathrm{mm}$) ?

- How do you deal with the non-gaussian behaviour of precipitation. Did you consider statistics based on ratios?

- Why do you use range statistics for all data and scatter plots with R2 for "hits only" data and not the opposite.

- Why some statistics are not shown in the figures (e.g. PoFD)?

- You only show the best and worst algorithms which exhibit huge performance differences making the figures difficult to read. There is hardly no information on the performance of the other 53 algorithms. You could show the best for each general type of algorithm.

- There is no comparison between the radars at gauges located at similar distances. This could highlight calibration issues.

- Why not showing the summer results for all radars?

**2.5 Are the results sufficient to support the interpretations and conclusions?**

In general the results are not sufficient to support the interpretations and there is a lack of discussion. It is particularly difficult to interpret the results when only the best and worst algorithm (out of 55) are shown.

You associate peaks in bias/error at specific ranges between 100 km and 150 km with bright band effects. I don't see why the overestimation due to bright band effect would not be spread over the range. The peaks appear strongly correlated with the falsely detected precipitation from the radar. In case of bright band the gauges should still measure precipitation. I think other radar errors or even gauge errors should be considered. Are the peaks caused by a few values? I would detect the largest errors and check visually for their cause. How are the results with "hits only" data?.

For all data the best and worst algorithm are R(Z,ZDR) and R(ZDR,KDP), respectively. This is interesting but not sufficiently discussed. What explains the bad performance of R(ZDR,KDP)? I am surprised that R(Z,ZDR) is the best since from the literature it is difficult to calibrate ZDR.

As expected, the seasonal results reveal a significantly higher contribution from summer precipitation to the statistics. There is no need to expand too much on this. However, a comment on the range performance is welcome. I don't see the winter data contributing more to PoFD than summer (line 312).

The results of Figures 10-12 for the "hits" data are interesting and allow some comparison between the algorithms. It clearly shows the added value of R(KDP) at long ranges. Again showing only the best and worst algorithm limits the interest. What is the performance of R(Z,ZDR) here? You don't discuss the bad correlation of some gauges which correspond to the peaks in the "all data" results.

**2.6 Do the authors give proper credit to related work and clearly indicate their own new/original contribution?**

The results of your previous study are not presented or discussed.

What is exactly done by WDSS? Also the rainrate estimation and the statistics?

**2.7 Is the description of experiments and calculations sufficiently complete and precise to allow their reproduction by fellow scientists (traceability of results)?**

The verification methodology is not clearly described :

- Some statistics are not defined and there are no references.

- The justification for NMB sounds inappropriate to me. You simply divide the mean bias by the sum of gauges values. Is it not to allow comparison between the gauges?

- The definition of the probabilities of detection between the radar and the gauge and related amounts are unclear. You are mixing probability and rainfall amounts in the explanation. Why would you scale the probability by the amount of precipitation? A reference from the forecast verification literature can help.

- You should already describe here how you select the data (all data or hits only).

- How do you define zeros in the data ? Do you apply any kind of thresholds? What are the implicit thresholds in the measurements?

The presentation of the results is unclear :

- How do you determine the overall worst and best algorithm?

- The text and figure caption are confusing. Which algorithm are shown in the figures 2-8? Best/Worst overall, per stats or per gauges? Also for PoD I guess?

- There are no units on the figure itself and the scale variations between the figures make comparison difficult.

- The text would be clearer if you compare the radar performance in one paragraph.

- Which algorithm is used on Figure 9?

**2.8 Does the title clearly reflect the contents of the paper?**

The title is not good:

- "Multi radar" suggests that a composite is used

- there is no reference to dual-polarization

- "at large ranges" is too limited (I would not mention range at all)

**2.9 Does the abstract provide a concise and complete summary?**

There are several issues :

- the introduction is too long

- the methodology is not well described

- the main results from the conclusions are not mentioned

**2.10 Is the overall presentation well structured and clear?**

The overall presentation is relatively well structured but the section and subsections titles are confusing (e.g Study area and methods, Individual radar performance).

**2.11 Is the language fluent and precise?**

The text is difficult to read :

- there is a lack of structure and logic at the level of the paragraphs (e.g. lines 53-58)

- the use of the terminology is sometimes inappropriate or lacks precision (e.g. line 68 and line 403)

- the style is relatively poor and inconsistent

- there are several grammatical and spelling errors

**2.12 Are mathematical formulae, symbols, abbreviations, and units correctly defined and used?**

The definition of the statistics and especially the probabilities are imprecise. The NMB formula seems incorrect (do you need to divide by N)? The acronym FAR (False Alarm Ratio) is used once on line 281 but not defined.

**2.13 Should any parts of the paper (text, formulae, figures, tables) be clarified, reduced, combined, or eliminated?**

In general the text should be rewritten to improve concision and clarity.

The first parts of the introduction (i.e. description of dual-pol technology) can be made shorter.

The "results and discussion" section should be reorganised. Parts of the methodology (e.g. selection of the data) is described in this section and should

be moved in the corresponding section. There are too much figures of the same kind. You analyse the 3 radars successively with a decreasing level of details. There is apparently no logic for that. I suggest to put the results of the 3 radars on the same figure and comment it in the same paragraph. I suggest to put the results of winter and summer data on the same figure.

The conclusions section should be reorganised. You don't have to repeat the literature review at the beginning. There is not a lot of general discussion with similar studies. The outlook at the end of the conclusions is too limited and general.

**2.14 Are the number and quality of references appropriate?**

The current literature review is limited and not well structured.

The review of existing verification studies of dual-pol radars is limited. While the operational polarimetric radar starts in 2012 in the US, there are not many references from the last years. The review is focused on the US while the dual-pol technology is available in other parts of the world, especially in Europe (e.g., Figueras i Ventura et al., 2012).

You mentioned only briefly the source of errors affecting all type of radars. However you refer to these errors often when you interpret the results (e.g. bright band). There is an extensive literature on the topic but you should at least cite a few key papers (e.g., Uijlenhoet and Berne, 2008).

You should also reference long-term verification studies of rainfall estimates from single-pol radars.

**References**

Figueras i Ventura, J., Boumahmoud, A.-A., Fradon, B., Dupuy, P., and Tabary, P.: Long-term monitoring of French polarimetric radar data quality and evaluation of several polarimetric quantitative precipitation estimators in ideal conditions for operational implementation at C-band, Quarterly Journal of the Royal Meteorological Society, 138, 2212–2228, doi:10.1002/qj.1934, 2012.

Uijlenhoet, R. and Berne, A.: Stochastic simulation experiment to assess radar rainfall retrieval uncertainties associated with attenuation and its correction, Hydrology and Earth System Sciences, 12, 587, doi:10.5194/hess-12-587-2008, 2008.

---

## Referee Comment (RC2) · Anonymous Referee #2 · 21 Sep 2017

Title: Multi radar performance in the Midwestern United States at large ranges Authors: M.J. Simpson and N.I. Fox

The current work presents the performance of 15 difference dual-polarimetric radar algorithms for 46 days of weather radar observations obtained by three radars in Missouri. Even though different algorithms perform differently for different events and different periods of the year, no clear conclusion can be drawn on which algorithm performs best. Even though some do seem to perform better than others. The general idea behind this study is very interesting and I enjoyed reading the presented results. However, I have the feeling this study is not finished yet and as such not ready for publication. The limited number of gauges as well as limited number of days make it difficult to draw any firm conclusions. Therefore, I would suggest that the authors would

do a number of additional analyses and incorporate a number of suggested changes to the manuscript. Once these are done, this manuscript will become very interesting. A complete description of my comments is given below.

Overall comments

Currently, only 46 days of precipitation are analyzed for a single year. I would suggest that the authors would extend this analysis covering multiple years to improve the robustness of the obtained statistics.

The current manuscript only makes use of a very limited number of rain gauges (15), which makes it difficult draw conclusions in general (especially for 46 days only). Many of the results presented in this work (Figures 2 – 8) present maxima and minima on solid lines, which gives the impression as if the radar performance is specific at a given range. Instead these maxima and minima are obtained for a given rain gauge only. I would suggest that the authors present the individual gauges in single points in these figures instead of solid lines. Next, in the presentation of the results, please be careful with generalizing certain phenomena that only hold for a single gauge.

No clear distinction was made between convective and stratiform precipitation. Even though the manuscript does indicate that convective precipitation is identifies (although using a very poor algorithm). It would be interesting if the authors could present there results for precipitation type especially in case the size of the dataset is extended (see previous point).

The authors currently present the results for each individual radar even though at the national scale these observations are merged into a single product. Therefore, besides presenting results as a function of distance from the radar, it would be valuable if the authors would generated a combined radar grid on which the performance of each algorithm would be calculated.

This manuscript misses a clear discussion about the impact of the results presented

here as well as the limitation of the applied methods. I would suggest that the authors would add this.

Specific comments:

Lines 46-48: The paper discusses the operational dual-polarimetric NEXRAD product. As the such, the X-band radars discusses in these lines fall outside the scope of the manuscript (as the operational S-band is not affected by attenuation) and should be removed as they do not add anything.

Lines 56-58: These lines are unclear. I would suggest that these are rewritten.

Lines 59-64: I would suggest to add the following references here: Kirstetter P.E., et al., 2013, JAMC, 52, 1645-1663, doi: 10.1175/JAMC-D-12-0228.1 and Hazenberg P., et al. 2013, JGR, 118, 10243-10261, 10.1002/jgrd.50726.

Lines 65-72: Quite a number of papers have looked during the last decade to the long-term performance of the operational weather radar. Though noted, these other papers might have not looked at a similar number of algorithms, which is something that makes this manuscript very interesting. I would therefore suggest that the authors rewrite this section given a bit more credit to work performed by others.

Lines 79-83: How does this work go beyond work presented by these authors in previous work? Even though 46 days is a nice amount, it only covers 1.5 month for a single year. Therefore, it is difficult to draw any firm overall conclusions especially when it comes to the impact of seasonality.

Lines 112-114: It is stated that 46 days of radar data were analyzed. Of the 46 days, how many hours did actually contain rainfall? As I suspect that it was not raining all day. If a considerable number of hours contained zero rainfall, what is the effect of this on the presented statistics?

Lines 118-120: This is a very classical approach which has been shown to be too simplistic. I would suggest that the authors make use of the method presented by Steiner M., et al., 1995, JAM, 34, 1978-2007, doi: 10.1175/1520-0450(1995)034<1978:CCOTDS>2.0.CO;2

Lines 141-143: How is the conversion from polar to Cartisian performed? Using the nearest point, or a weighted integration?

Lines 143-145: While integrating the 5-minute precipitation product to hourly intervals, was there any spatial interpolations between individual images performed? Especially, for summertime convective summer events, not accounting for the propagation speed of a precipitation cell between individual 5-minute scan can have a serious effect on the hourly accumulations (see e.g. Fabry et al., 1994, JoH, 161, 415-428, doi: 10.1016/0022-1694(94)90138-4). In case this was not taken into account, how would this potentially affect the obtained results presented here?

Lines 145-147: Why would it not be possible to use the nearest polar pixel for comparison with the rain gauge?

Line 75 and lines 154-155 state that a total of 55 algorithms were applied while in lines 156-160, a total of 15 methods are briefly presented. Please clarify this difference.

Lines 217-222: The authors suggest that the overestimation by the radar at around 150 km might be due to the bright band. First it is not clear what is meant with a "second bright-band". Instead of suggesting the possibility of bright-band contamination, I would suggest that the authors analyze local sounding/weather model observations for the different days analyzed to obtain a proper estimate of the location of the zero-degree isotherm and at which distance the radar beam interaction with the layer just below this. This will help to clarify whether the maximum was indeed related to bright band. Next, I would suggest also to carefully look at the convective/non-convective data, as the former should not be affected by the bright-band.

Lines 322-326: What about the fact that winter precipitation is generally more frontal and widespread with spatially variabilities much smaller. As compared to summer precipitation with convection triggering small-scale variations. As such, it is easier for the radar to see the proper evolution of precipitation in winter as compared to the summer, although correlation coefficient does not indicate the performance to estimate the exact amount.

Lines 337-349: Given the limited number of gauges used in this study, I would suggest that the authors would be careful to make any subdivisions with respect to distance.

Figure 9: Which radar product is being used here?

Figure 10: It makes statistically to identify the impact of a given radar algorithm while looking at an individual gauge. Especially in case you only have 15 gauges.
* * *

---

## Referee Report (RR1)

**Comments to "RANGE AS A FUNCTION OF DUAL-POLARIZED QUANTITATIVE PRECIPITATION ESTIMATION"**

Edouard Goudenhoofdt

February 5, 2018

**1 General comments**

I would like to thank the authors for considering my comments and their responses. Unfortunately most of the major comments have not been addressed in a satisfactory manner. While clarifications are appreciated, some of the responses are missing, irrelevant or too short. There are no clear references to the revised manuscript, which does not contain the major changes expected. Additions to the text are made but limited to a few sentences which are often not well integrated. The new figure comparing the performance of all algorithms reveals the potential of the paper which is still not fully exploited. Below is a summary of the most important remaining issues.

**2 Major comments**

**2.1 Does the paper address relevant scientific questions within the scope of HESS?**

The paper could provide a long-term verification of dual-pol QPE algorithms which is relevant for hydrology. The authors stress that they focus on the range effect but this is in contradiction with the extended list of objectives in the introduction and the limited amount of results related to range in the conclusions.

**2.2 Does the paper present novel concepts, ideas, tools, or data?**

The number of data is limited. Why only one year? Why only 46 days of precipitation are available when the normal is around 100 days?

**2.3 Are substantial conclusions reached?**

The conclusions are short and do not summarize clearly the main findings (i.e. the algorithm's relative performance in function of the range). A proper discussion on the validity and possible cause of the different results is missing.

**2.4 Are the scientific methods and assumptions valid and clearly outlined?**

The information on the data and their quality is still limited while it seems some observation errors affect the results. Which type of quality control is effectively performed by WDSS-II on the radar data? Why not using the one-hour precipitation product of NOAA as reference? Why using the Mesonet network when the higher resolution CoCoRaHS is considered as better by the authors? The data selection criteria and choice of statistics are not sufficiently discussed.

**2.5 Are the results sufficient to support the interpretations and conclusions?**

In Figure 2, the results vary a lot between the algorithm's and the radars making interpretations difficult. I am surprised by the bad performance of KDP (did you check the cause visually?). The tentative explanations of radar issues for specific gauges (e.g. bright band effect) are not robust. In Figure 3-8, only the overall best and worst algorithm's are shown, which is too limited (I would present the best of each type). It is often unclear for which algorithm an interpretation is valid. There are inconsistencies in the results : the NSE is sometimes different between Figure 2 and the other figures ; the overall results are not always equal to the sum of the cool and warm seasons results.

**2.6 Do the authors give proper credit to related work and clearly indicate their own new/original contribution?**

The results of similar studies (including from the authors) are not properly reviewed. Is there a connection with your recently submitted article on X-Band?

**2.7 Is the description of experiments and calculations sufficiently complete and precise to allow their reproduction by fellow scientists (traceability of results)?**

The description of the statistical analyses needs to be much more clear and precise (proper definition and interpretation, thresholds used for zeros, selection of hit only data).

**2.8 Does the title clearly reflect the contents of the paper?**

The new title sounds a bit odd to me.

**2.9 Does the abstract provide a concise and complete summary?**

The abstract has not been improved as suggested and is not consistent with the conclusions.

**2.10 Is the overall presentation well structured and clear?**

The comments have not been taken into account. There is still part of the methodology in the "results" section.

**2.11 Is the language fluent and precise?**

No significant efforts have been made to improve the text structure, terminology and style. There are annoying editing errors at this stage (e.g. a repeated sentence on line 212).

**2.12 Are mathematical formulae, symbols, abbreviations, and units correctly defined and used?**

Some definitions are still incorrect or imprecise.

**2.13 Should any parts of the paper (text, formulae, figures, tables) be clarified, reduced, combined, or eliminated?**

The results section is still not clear nor concise. There are too much points in Figure 2. There are too much plots in the figures. I would show only NMB, NME, PoFD, PoD. Paragraphs over the different radars could be combined. What is exactly on figures 2-8 : best at each point (your response) or only R(Z,ZDR) (figure caption)?

**2.14 Are the number and quality of references appropriate?**

The number and quality of the references are acceptable but they are often cited for anecdotal reasons (e.g. Figueras et al. on line 381). They are best used for discussion in the introduction and conclusions sections.

---

## Author Response (AR2)

Reviewer 1 Comments and Response:

**Compared to the previous version, this updated version of the manuscript has improved a lot. Although I should note that not all suggestions raised have been accounted for. However, the performed work, results and conclusions are well presented.**
**There is only one final detail that I would like to see altered before I feel the manuscript is ready for publication. At multiple places (both in the abstract and conclusion) the authors talk about 1100 of radar precipitation observations. However, this is just 46 days of data. At another location details are provided that actually 400 of the 1100 hours contained precipitation. I would therefore suggest that the authors alter the 1100 into 400 hours of precipitation**

We thank the reviewer for the above comments. We have updated the necessary changes throughout the manuscript to properly reflect the correct number of days and precipitation amounts.

Reviewer 2 Comments and Responses:

**The paper could provide a long-term verification of dual-pol QPE algorithms which is relevant for hydrology. The authors stress that they focus on the range effect but this is in contradiction with the extended list of objectives in the introduction and the limited amount of results related to range in the conclusions**

We appreciate this comment. We have added discussion in the conclusions to elaborate upon this aspect. We also added elaboration in the list of objects (near line 79 on page 3) to emphasize the range effect.

**The number of data is limited. Why only one year? Why only 46 days of precipitation are available when the normal is around 100 days?**

We chose a random year for the analyses to be conducted, we elaborated that 100 days have 'measureable' rainfall (i.e., greater than trace) whereas 50 days have greater than 0.5mm in of rainfall. Therefore, the 46 days chosen / analyzed falls near the average amount of days with appreciable rainfall.

**The conclusions are short and do not summarize clearly the main findings (i.e. the algorithm's relative performance in function of the range). A proper discussion on the validity and possible cause of the different results is missing**

Thank you for this comment. We have expanded upon the conclusions which were lacking in wrapping the paper up.

**The information on the data and their quality is still limited while it seems some observation errors affect the results. Which type of quality control is effectively performed by WDSS-II on the radar data? Why not using the one-hour precipitation product of NOAA as reference? Why using the Mesonet network when the higher resolution CoCoRaHS is considered as better by the authors? The data selection criteria and choice of statistics are not sufficiently discussed.**

We have added a more detailed description of the quality controlled techniques implemented, which would mitigate large errors in QPE from various modules within the WDSS-II framework. We did not consider using the DP rate as a reference, as that is more of a heuristic algorithm that blends multiple different algorithms together (it is difficult to determine whether they implement R(KDP), R(Z,ZDR), etc.) without doing a deep analysis of the radar parameter values as well as the particular algorithm implemented at each time. Furthermore, it is difficult to determine whether each of the 3 radar locations implement the same sort of dual-pol radar equation at the same times. Lastly, We implemented the Mesonet data due to the timing in which the current study was conducted. The authors have follow-up studies which utilize CoCoRaHS, HADS, MADIS, and other gauge locations.

**In Figure 2, the results vary a lot between the algorithm's and the radars making interpretations difficult. I am surprised by the bad performance of KDP (did you check the cause visually?). The tentative explanations of radar issues for specific gauges (e.g. bright band effect) are not robust. In Figure 3-8, only the overall best and worst algorithm's are shown, which is too limited (I would present the best of each type). It is often unclear for which algorithm an interpretation is valid.**

We thank the reviewer for these comments. After checking visually, bright-banding were present in several cases, but the w2qcnndp as well as w2qcnn algorithms *should* have handled them effectively (cases slip through, of course). We have addressed this within the text which is primarily the result of the large biases observed in spite of larger distance from the radar. The algorithms represented via Figures 3-8 are labeled within the caption and represent the best-performing R(Z,ZDR) and worst performing R(ZDR,KDP) algorithms. This helps to highlight differences between the algorithms not only between the warm, but also the cool season.

**The results of similar studies (including from the authors) are not properly reviewed. Is there a connection with your recently submitted article on X-Band?**

We have seen similarities with the superiority of R(Z,ZDR) algorithms over R(ZDR,KDP) or R(KDP). We did, as well, see superiority with the R(Z)-Convective equation as well.

**The description of the statistical analyses needs to be much more clear and precise (proper definition and interpretation, thresholds used for zeros, selection of hit only data)**

We appreciate this comment, and have elaborated on the definition of thresholds and hit only at the end of the statistical analyses section in which more than 2 tips were needed for calculations to be implemented.

**The new title sounds a bit odd to me**

We have changed the title of the article to make it flow better.

**The abstract has not been improved as suggested and is not consistent with the conclusions.**

We appreciate this reviewer comment, and have expanded on the abstract to better reflect the conclusion, make it easier to read, and fixed some spelling errors.

**The comments have not been taken into account. There is still part of the methodology in the "results" section.**

We thank the reviewer for this comment, and we have moved text to/from the methodology and results section to better reflect the text within each section.

**No significant efforts have been made to improve the text structure, terminology and style. There are annoying editing errors at this stage (e.g. a repeated sentence on line 212)**

We have moved text around throughout the methodology and results to create a better-flowing manuscript.

**Some definitions are still incorrect or imprecise**

The authors thank the reviewer for this comment. We have gone through the text and ensured accuracy in the definition and spelling of each acronym.

**The results section is still not clear nor concise. There are too much points in Figure 2. There are too much plots in the figures. I would show only NMB, NME, PoFD, PoD. Paragraphs over the different radars could be combined. What is exactly on figures 2-8 : best at each point (your response) or only R(Z,ZDR) (figure caption)?**

We have utilized only the best algorithm from the set of R(Z,ZDR) equations and the worst algorithm from the set of R(ZDR,KDP) equations as these consistently showed to be the best and worst, respectively. We have implemented the data from each of the statistical analyses to better represent the performance of each algorithm at each radar. Some results would not have been accounted for or even could have been completely missed without some of the statistical measures analyzed in this fashion.

**The number and quality of the references are acceptable but they are often cited for anecdotal reasons (e.g. Figueras et al. on line 381). They are best used for discussion in the introduction and conclusions sections.**

We have altered our references and moved them around to be more appropriate for the current study.

[revised manuscript text omitted]

---

## Author Response (AR3)

For final publication, the manuscript should be
**accepted as is**
**accepted subject to technical corrections**
accepted subject to **minor revisions**
reconsidered after **major revisions**
    I am willing to review the revised paper.
    I am **not** willing to review the revised paper.
**rejected**

**Suggestions for revision or reasons for rejection (will be published if the paper is accepted for final publication)**
In their response the reviewers, the authors state that "50 days have greater than 0.5mm in of rainfall.", while in text it is stated in Section 2.2 "with only 50 days typically recording greater than 25.4 mm". As this latter intensity value is quite sever, please make sure this is not a typo.

We thank the reviewer for catching this oversight. We have made the necessary change on line 115 of the text. We have also made several small grammatically changes through the text.